# The Outbreak of Digital Detox Motives and Their Public Health Implications for Holiday Destinations

**DOI:** 10.3390/ijerph19031548

**Published:** 2022-01-29

**Authors:** Gonzalo Díaz-Meneses, Miriam Estupinán-Ojeda

**Affiliations:** Department of Economics and Business, University of Las Palmas de Gran Canaria, 35017 Las Palmas de Gran Canaria, Spain; miriam.estupinan101@alu.ulpgc.es

**Keywords:** digital detox, tourist behavior, marketing

## Abstract

This paper aims to analyze the external and objective barriers of the digital difference between being at home and being on holiday, and the intrinsic and subjective inhibitors to remaining online once at a destination. In this study, the literature is thoroughly reviewed, going beyond the traditional economic and technological explanations, along with those related to skill, to consider those rooted in well-being and psychology. Hence, a more integrative and exhaustive framework deals with how tourists approach their perceived hazardous and oversaturating digital environment. Finally, the role played by sociodemographics is studied by profiling those who are predisposed toward disconnecting in order to preserve their wellness. In total, 346 tourists were surveyed at random, with proportional stratification, on the island of Gran Canaria. The measuring instrument comprised a questionnaire whose scales gathered information about more than eighteen devices, twenty-eight social media platforms, and sixteen device and social media barriers. The obtained evidence demonstrates how crucial “detox” motivations are when trying to elucidate the differences in digital behavior between their home and holiday destination. Similarly, the evidence highlights that while gender, age, nationality, and income are associated with these differences, education is not. This study pioneers an analysis of the detox barrier regarding staying connected while on holiday and provides insight into how this intrinsic and subjective inhibitor interacts with other external hindrances to people’s health, both where they live and where they travel.

## 1. Introduction

There is a difference between the mindset of daily life and the one on holiday, where there is a desire for a change of pace and a break from routine [1], as well as an array of additional factors that come into play, such as the desire to have a unique experience and experience freedom from everyday demands [2]. In addition to these factors, when on holiday there are mental barriers to going online [3] and difficulties with technology [4], causing differences in behavior compared with that at home. These differences affect the rate at which visitors use their electronic devices and determine significant digital differences between online behavior at their place of origin and destination, the rationale of which lies in discretionary consumer behavior [3,5]. In fact, there exist multiple digital differences that have received little attention in tourism research, mainly in relation to tourists coming from countries with a high level of online access [6,7]. However, the emergence of mobile devices has lessened this separation [8,9] and undermined the “escape” motivation [1,5,10]. Not only are there concerns about how abundant information and multiple tasks affect people’s well-being [11,12] but it also represents a research gap [13,14]. These differences undoubtedly stem from health disparities and are relevant for public health. For this reason, and in line with much of the literature that associates tourism with quality of life [1,15], well-being [16], and health [17,18,19,20,21], the current paper brings into focus the specific line of research consisting of the digital disparity between origin and destination and the dimension that regards, essentially, not only physical access, use, and skill differences but also a purposeful, motivational, emotional, intrinsic, and subjective discrepancy.

Firstly, the attention of researchers has shifted away from infrastructural-technological barriers toward skill and usage hindrances [3]. Secondly, new inhibitors have been put forward that lie beyond objective obstacles, such as lack of trust [6,22], the pursuit of relaxation and self-serendipity [23], and a broad variety of psychological, intrinsic, and subjective constraints [24]. However, whereas the former has been sufficiently analyzed using economic, technological, and socio-capability models [6], the latter has been less well-researched; thus, the theoretical framework should be broadened [7,25,26] toward integrating an approach based more on psychology and health [27,28,29]. After all, traveling encompasses a wide set of motives, ranging from relaxation and stimulation to relationship maintenance, self-esteem, and growth [5]. On this basis, the first objective consists of analyzing not only external and objective inhibitors but also the intrinsic and subjective motivational barriers of digital differences related to the need for escape, stress relief and well-being. With this aim in mind, we touch upon the various theories that might give an account of the digital difference between home and holiday destination, and, moreover, the existing barriers to going online on holiday (see Figure 1).

Consequently, exactly how the influence of sociodemographic features operates in the wider sense on the digital difference presents a need for further research [30,31]. In fact, the emphasis in the existing research has been on the sociodemographic description of the digital divide [32,33], but these characteristics have not been associated with either psychosocial or psychological causes [3]. For this reason, the term “sociodemographic” has not been interpreted eclectically, notwithstanding the fact that there is room for this new approach if the existing literature is considered as the starting point for making new doctrinal inferences, for example, regarding wellness, quality of life, and security. To be specific, if gender can be considered as a shaped biological and cultural propensity to feel for and respond to new technologies [34], and age is a determining variable in being a “digital native” or “digital immigrant” [35,36], then it seems possible to provide more insight into their antecedent functions for generating digital differences, ranging from technology and literacy to security, privacy, and health tourism. Similarly, insofar as nationality is understood as an antecedent condition of an “info-structural” national environment [37], its applicability to the field of tourism—by making an allowance for researching its implications in terms of quality of life between origin and destination—is clear. Finally, if education is explained as a source of knowledge and skills [38], and income results in superior technological capacity or inequality [39], there exist psychosocial and power-related significances that have an impact on health that should be further studied in the field of tourism. On this basis, the second objective amounts to describing the sociodemographic profile of the digital difference between origin and destination, by considering not only external and objective hindrances but also intrinsic and subjective volitions. This sociodemographic approach allows seeking solutions, for example, against toxic digital influences, owing to the visible nature of gender, age, etc.

To achieve these objectives, the paper is divided into a review of the literature, methodology, an analysis of the results, and our conclusions.

## 2. The Review of the Literature

There are two types of barriers that explain the different digital behaviors of tourists between home and holiday destination that might be traditionally distinguished: firstly, infrastructural barriers, such as the lack of access to devices and the absence of opportunities to use them; secondly, usage barriers, such as the deficiency in elementary experience on digital platforms and a low level of digital skill [30]. In this respect, while the former used to be more important and was often attributed to objective inhibitors, the latter is gaining more recognition as a sort of subjective self-hindrance. In addition, there exists a kind of digital difference between home and holiday destination that goes beyond the merely economic, technological, and skill disadvantages, to account for the motivational barriers existing on holiday. It boils down to a disparity related to well-being and motivation and, hence, points at a psychological root cause [40].

Theoretically, and according to the causal model of resources [30], time and material resources are assets whose possession—or scarcity thereof—account for the existence of these digital disparities. Similarly, the economic and technical approaches explain insufficiencies in access and use [6]. Likewise, the sociotechnical, as well as the socio-capability models, shed light on the engagement in terms of ICT [6,16]. Nevertheless, a more integrative and comprehensive framework is necessary in order to include more net motivational factors [6]. These volitional elements are particularly relevant in the field of tourism, where the inhibitors and lack of motivation might not only account for a significant existing difference in digital behavior at both their point of origin and destination but may also be rooted in psychological, spiritual, and completely subjective factors. In this vein, the theory of systems for well-being addresses the idea that the barriers to connecting virtually at a destination may not only come down to destination infrastructure, market trends, and finance related to business and networks but also to brand communication and how tourists interpret it [16].

In this regard, the appropriation theory might explain how there are mental, social, and cultural assets whose adoption could clarify why some people do not use digital technologies on holiday. For this framework, people might reject digital experiences on holiday if they are not meaningful for them [40], for example, because it is constructed around the point of origin and work rather than the destination. By the same token [41], we can identify barriers by inversely interpreting the TAM model. Consistently, the rejection of connectivity might be explained not only by external variables, the lack of ease of use, and the perceived pointlessness in utility but also by the negative attitude toward technology, the unperceived mobility value, and even feelings that digital activities are tiresome. Going by the travel career ladder, it is not disconnection that matters most, but rather how repeat visitors learn the best way to meet their needs at a destination, thanks to past experience [5]. The suggestion is proposed by the theory of travel career patterns that once tourists have had the experience and move up in the pyramid of motivation, they are driven by fulfillment and fewer detox needs [5]. Similarly, other models acknowledge the effects from both internal and external variables by stating that the usage of technology is shaped by society, history, and institutions, such as are proposed by the theory of affordances [42]. There is no doubt that destinations attract visitors by pulling and pushing on internal and external forces [5].

At this point, it is worth mentioning the emergence of a new niche in research regarding anxiety and stress caused by frequent online activity, highlighting the apparent and increasing demand for “digital detox” holidays, consisting of being disconnected from all social media and online activity [15]. Therefore, the disconnected emotions model gives credence to the emotions felt about technology [42] and gives an account for technostress [43,44], pointing out that social media and smartphones enhance a degree of social presence that traps people in work communications and, in this way, limits the recovery experience. Similarly, it may be pointed out that there must be people with outstanding digital resources and skills but with the determination to limit the use of the new technologies in their daily lives because, for example, they perceive a high degree of personal insecurity [22]. Likewise, a notable amount of the literature brings into focus the relevance of more subjective barriers to the use of digital technologies, such as the desire to enjoy nature and social relationships in person and without distraction [8], the fear of being the victim of online fraud and scams, and the need to escape and restore oneself from the demands of work and its intrusions into free time [1,44]. Equally, we can apply the concept of e-lineation to the field of tourism and refer to the search for authenticity on the part of tourists, in their aspiration to self-realize in conjunction with their social and physical environment [45]. Needless to say, that alienation resembles the separation between the artificial constituency of work and the ideal human nature [45]. On this basis, Hypotheses 1 and 2 are put forward, as follows:

**Hypothesis** **1** **(H1).***The different digital behaviors of tourists between the point of origin and destination are associated with external and objective barriers*.

**Hypothesis** **2** **(H2).***The different digital behaviors of tourists between the point of origin and destination are associated with both intrinsic and subjective barriers*.

Personal characteristics determine the resource allocation and, in turn, differential access to and different appropriation of technology [40]. In this respect, cultural resources are interpreted differently in accordance with gender roles, where performance influences the technological usage to identify and represent, mostly, a sense of masculinity or femininity [33,46,47]. In addition, insofar as the availability of time is a key resource for using technology, and in terms of there being a generational approach to explaining how every cohort uses technology, age is a determining factor for profiling the different digital behaviors in the field of tourism [30,48,49]. Equally, nationality refers to a social context in which people develop their network and handle social media; therefore, digital behaviors are differently reliant on nationality and national culture [22,50,51]. Likewise, it stands to reason that resources stemming from power, ability, and motivation are logically associated with the level and context of education [52]. Finally, as another resource consists of affordability and the level of quality of life sourced by wealth, income undoubtedly accounts for differences in terms of control over personal experiences, and so in terms of digital inequalities [53]. On this basis, Hypotheses 3 and 4 are put forward as follows:

**Hypothesis** **3** **(H3).**
*The sociodemographic profile of tourists explains the differences in external and objective barriers to using social media at a destination.*


**Hypothesis** **4** **(H4).**
*The sociodemographic profile of tourists explains the differences in intrinsic and subjective barriers to using social media at a destination.*


There is a glaring difference in the patterns of socialization of males and females that contributes toward the difference in digital behaviors at the destination. According to the theory of cultural socialization roles [54], there are dissimilar usage patterns for digital devices and social media, depending on gender, the outcome of which is a lesser perception of entertainment value in females and a more favorable attitude toward information technology (IT) in males. This stereotypical disparity determines that women feel less comfortable and display a “lower-skilled” self-concept, and so are more prone to feeling anxiety and suffering from “computer phobia” [33], although some authors claim there would be no differences if people fulfilled the same roles [54]. Nevertheless, the theory of attribution states that men and women develop two different attribution models, leading men to ascribe their successful performance in the digital field to their intrinsic ability, while women tend to credit it to the effort they put in or to luck [33]. Similarly, the theory of role constraints points out the existence of different coping strategies determined by enforced gender socialization, the consequence of which is a higher probability of “psychological nuisance” in women than in men [54]. Finally, a broader interpretation of inequality theory might lead one to assert that women are undermined by different social positions, with unfavorable power circumstances that eventually relate to a higher number of stressful and frustrating situations and, hence, greater detoxing needs [55]. In this vein, women exhibit a higher frequency of stressors when traveling [56], due to their stress vulnerability profile [57]. For example, while women are prone to suffering more stress from social anxieties, men are more likely to be affected by task overload [58]. Afterward, given that anxiety and depression are more likely to be reported by women, one might argue the crucial importance of learned cognitive and emotional patterns. In other words, there is a key role played by interiorized interpersonal behaviors that are shaped by culture [57]. All these explanations configure a kind of digital behavior at the destination related to gender, showing distinct patterns.

On this basis, research on digital literacy has assumed that women are less oriented toward technology and, hence, are more sensitive to any adverse digital circumstances [40]. However, this disparity is due to the existence of different skills and different ways of using technology that may lead women to use digital devices and the internet less frequently [30]. In fact, when the level of risk, perceived insecurity, and sensitivity about privacy are higher, the probability of making purchases online is lower in women [34]. This results in a greater need for disconnection and a more significant desire for “digital detox” holidays in women than in their male counterparts [1]. Consequently, Hypotheses 3a and 4a are put forward, as follows:

**Hypothesis** **3a** **(H3a).***The gender of tourists explains the differences in external and objective barriers to using social media at the destination*.

**Hypothesis** **4a** **(H4a).**
*The gender of tourists explains the differences in intrinsic and subjective barriers to using social media at the destination.*


It goes without saying that there is also a different digital behavior related to age that distinguishes the way people handle not only technology but also social interactions [31]. To be specific, the predominant literature states that the oldest segment of the population shows some reluctance to adopt new technology [34], although this difference is moderated by education and income [36]. However, the relationship between the use level of digital technologies and age is exponential [36] but not linear, since the age of fifty was considered a turning point for e-literacy—although, possibly, this limit will disappear in the future [38]. In contrast, young cohorts are more inclined to use technology. In any case, the barriers reported by older users stem from fraud risk apprehension, threats against privacy, and a lack of digital skills [36]. Nevertheless, age is not an intellective criterion and, therefore, it does not predispose one toward disconnecting, although younger users tend to have more advanced mobile devices, the immediate effect of which is that disconnection is less likely [36]. What is more, while younger people are more predisposed toward overusing technology because of a fear of “missing out” [59], nomophobia behaviors [60], and additions to a smartphone [61], older people tend to feel saturated with information [62].

The range of theoretical approaches to account for digital age differences at a destination is diverse. Firstly, from the social capital theory standpoint, it can be argued that younger users show more digital literacy, self-efficacy, and social support than older people, whose age is closely related to a broad array of extra learning costs [36]. Secondly, the application of “generation theory” to tourism has demonstrated significant differences in terms of possession, usage, ability, and motivation. To be specific, it addresses a glaring difference between “millennials”—the cohort with the highest level of digital literacy—and the remaining generations, whose shortcomings are related to different life experiences with digital technologies. For example, the usage of mobile technologies was adopted, with effort, by “generation X”, and, rather late in their adulthood, the internet was tentatively embraced by the vast majority of “baby boomers”, although digital devices are still challenging the capabilities of “the silent generation” [63]. Consistently, generation theory addresses a disparity between “digital natives” and “digital immigrants” and points out that while the former make less effort and enjoy a wider variety of devices and online experiences, the latter resist the new IT tools and show fewer digital competencies [15]. Hence, one might state that digitally native millennials handle their potential stress better than their counterparts from generation X, baby boomer, and silent generation cohorts [64].

In addition, it is worth stating generation theory might also be put forward in order to associate each cohort with different tourism purposes. To be precise, it seems evident that baby boomers, more than any other cohort, are the fondest of wellness tourism and the need for relaxation over any other motive. Similarly, there exist more health motives to travel [65] and self-assessed health considerations [66], in older than younger people. Hence, one might argue that alleviation from stressful circumstances is the key objective for this particular generation when taking a break [67] and, thus, they are more predisposed than other, younger cohorts to choosing holidays where they can disconnect, detox, and pursue activities related to health and well-being. After all, not only do the older cohorts experience more digital barriers but they also face more health issues [68] and, consequently, exhibit a different pattern of traveling [64]. On this basis, Hypotheses 3b and 4b are put forward:

**Hypothesis** **3b** **(H3b).***The age of tourists explains the differences in external and objective barriers to using social media at a destination*.

**Hypothesis** **4b** **(H4b).***The age of tourists explains the differences in intrinsic and subjective barriers to using social media at a destination*.

The distinct digital behaviors at holiday destinations might be the result of differences in terms of internet usage that might be traced back to differences in how economics, regulation, and politics have evolved over the past few decades [37]. Nevertheless, people and organizations are not only affected by non-controlled environmental factors but are also active subjects in their current situation [69]. In this sense, within the European Union, there are significant differences between countries in respect to the level of internet access and digital literacy and, in turn, to their respective citizens being more or less prone to disconnecting during holidays, the south being generally less developed and more likely to disconnect than the north [70]. A theoretical framework to account for these differences is the model of innovation diffusion, whose application to geographical context is quite suitable and extensive [38]. By using this model, one might find grounds to explain why the most advanced countries suffer fewer objective barriers than in the less advanced countries. As with other economic and technical models, the emphasis is put on the degree of digital advancement existing in a particular market and how quickly the life cycle of technology develops.

Nevertheless, as long as a less sociotechnical approach is adopted, one might explain differences regarding subjective barriers. It is worth noting that technical and social models account for the existence of differences between “digital native” countries and “digital non-native” countries [35,64,71,72]. While the former comprises objective variables, such as infrastructure, utilization and opportunity, the latter includes more subjective criteria, such as cognition and motivation. For example, the rainbow model highlights the importance of the population’s skills and literacy, social facilitation, and the political framework [6]. Likewise, social capabilities models bring into focus the existence of different digital behaviors and the possibility of explaining the susceptibility degree of engagement in new technologies by communities, as well as the corresponding different motivational levels to having, using, learning about and enjoying the internet [73].

However, by making use of the cross-cultural national approach, one might talk about “male” and “female” countries [74], so that different digital behaviors at a destination between different nationalities might be further examined. On this basis, different levels of inherent naturalism might be easily related to a nationality, and, thus, to a particular propensity toward suffering from external and objective or intrinsic and subjective inhibitions might be drawn discriminatorily as a conclusion [1,15,75]. To be specific, it seems coherent to assume that female-oriented and “uncertainty-sensitive” countries such as Spain and the Nordic countries will be more likely to express “detox” and “apprehension” motives than male-oriented countries, such as Germany and Holland. Consequently, Hypotheses 3c and 4c are put forward:

**Hypothesis** **3c** **(H3c).**
*The nationality of tourists explains the differences in external and objective barriers to using social media at a destination.*


**Hypothesis** **4c** **(H4c).***The nationality of tourists explains the differences in intrinsic and subjective barriers to using social media at a destination*.

People who are more well educated are more prone to using technology in their everyday lives and are more inclined toward making purchases online [34], notwithstanding that the predictive power of this variable tends to be lower [38]. Logically, education correlates strongly with the level of general skills and digital literacy, and, hence, offers a significant explanation for the existence of any different digital behavior that prioritizes the usage rather than the possession of skills [30]. Obviously, the tourism field reflects these disparities, so that less well-educated tourists are more likely to fall into the category of infrequent or “low-level” users [76].

Nevertheless, these pieces of evidence can be grounded on the theory of power and exposure and used to point out that less well-educated people are more likely to be exposed to negative lifestyle contexts [77], without neglecting completely the theories above based on resources and capabilities, the approach of which sheds light on the crucial role played by cognitive and practical skills [78,79]. In fact, the literature has empirically demonstrated that less well-educated people tend to have worse job conditions [80], and these conditions cause them to suffer from higher levels of digital saturation, due to their greater exposure to “toxic” digital situations [81,82]. On this basis, Hypotheses 3d and 4d are put forward:

**Hypothesis** **3d** **(H3d).**
*The education level of tourists explains the differences in external and objective barriers to using social media at a destination.*


**Hypothesis** **4d** **(H4d).***The education level of tourists explains the differences in intrinsic and subjective barriers to using social media at a destination*.

Income and education are significantly correlated, and this is the reason for highlighting the same theories of power and exposure. It is also true that income explains the different digital behaviors at a destination specifically in terms of purely economic inequality [30], although its applicability to the welfare context, where the level of economic disparity is lower, does not appear to be clear. In this sense, affordability based on relative wealth explains not only the availability of devices and access to IT infrastructure [32] but also the exposure to better status conditions. In fact, income loses its antecedent power if prices related to the possession of devices and access to the internet go down [38]. Therefore, income also works indirectly by affecting other variables related to the level of connectivity, such as social class, networks, and neighborhood opportunities, and even high-quality and self-realizing jobs [32,39].

Nevertheless, these pieces of evidence do not deny these theories loosely, based on a pure resource and capability content, the framework of which highlights the importance of affordability and access to a sociotechnical infrastructure [83,84,85], but instead give more credit to the applicability of explanations stemming from power and exposure [86]. In fact, people with a higher income show a higher proclivity toward making purchases online by appreciating the offered time savings, convenience, and value [34]. On this basis, Hypotheses 3e and 4e are put forward:

**Hypothesis** **3e** **(H3e).**
*The income of tourists explains the differences in external and objective barriers to using social media at a destination.*


**Hypothesis** **4e** **(H4e).***The income of tourists explains the differences in intrinsic and subjective barriers to using social media at a destination*.

## 3. Materials and Methods

A questionnaire was given at random to 346 subjects with proportional stratification at the beginning of 2015, on the island of Gran Canaria. Semi-structured scales of more than eighteen devices, twenty-eight social media platforms, and sixteen device and social media barriers—among other variables—are used as measuring instruments.

The measuring instruments employed in the questionnaire are as follows:Measure of devices: two different questions, each consisting of 18 items—on three-point classification scales—that asked the respondents to indicate the extent to which they use certain electronic devices at home and on holiday (see Figure 2).Measure of generic social media: two different questions comprising 13 items—on three-point classification scales—that correspond to the most well-known social media platforms, in order to gather information about their use at home and on holiday (see Figure 2).Measure of touristic social media: two different questions, each consisting of 15 items—on three-point classification scales—referencing tourist websites and applications that offer the possibility to share and participate online from both their point of origin and destination (see Figure 2).Measure of barriers: one question including 16 items—on a five-point Likert scale—that asked the respondent to choose a device and social media platform that they use at home but could not or did not want to use at their holiday destination. This scale is inspired by numerous research works [24,36,43,45,59,87,88].Measure of sociodemographic characteristics: several types of questions are formulated depending on the target feature. For gender, it is dichotic; for age, a classification scale with 5 points is used; for nationality, a semi-open-ended scale with five different nationalities; for education, one item scale with five points; and, for income, one item scale with four points.

## 4. Results

### 4.1. Preliminary Analysis

The disparity in the level of usage for devices and social media between home and a holiday destination is measured by subtraction, so that three new variables with information regarding the metric difference are created. To be specific, the first difference variable is called “devices”, the second, “general social media”, and the third, “tourist social media”.

In addition, exploratory factor analysis was performed in order to find out the dimensionality of the barriers to using devices and social media at a holiday destination, and four factors are extracted (see Table 1).

The first factor regards external reasons and objective factors about infrastructural, social and time constraints against using social media at the destination, and so is labeled “external and objective barriers to social media usage”. The second factor relates to apprehensive causes, such as fear that the device might be stolen, lost, or broken, and consequently is called “intrinsic and subjective barriers for devices”. The third factor refers to stress and the need to disconnect from electronic devices and social media and thus is named an “intrinsic and subjective detox barrier”. Finally, the fourth factor alludes to a variety of specific technical problems associated with devices, such as incompatibility with the destination’s infrastructure, excess of baggage and “does not work”, and, as a result, is termed an “extrinsic and objective barrier for devices”.

### 4.2. Statistical Analyses to Test Hypotheses 1 and 2

A correlation analysis was carried out in order to measure how closely associated the barriers to using devices and social media are to the different digital behaviors between their point of origin and holiday destination.

As is laid out in Table 2, the major barrier to staying connected is the external and objective barrier for social media usage since it shows a significant relationship with the level of general social media, tourist social media, and device practice. Secondly, the intrinsic and subjective detox barrier explains the difference in social media usage at the destination. Nevertheless, it is worth mentioning that the apprehensive intrinsic and subjective barriers for devices do not relate to any type of different digital behavior between home and the destination and, hence, Hypothesis 2, which states that the different digital behavior of tourists between origin and destination are associated with intrinsic and subjective barriers is only partially accepted. Lastly, the different digital behavior is associated with the extrinsic and objective barrier for devices. Therefore, Hypothesis 1, which proposes that the different digital behavior of tourists between origin and destination are associated with external and objective barriers is accepted.

### 4.3. Statistical Analyses to Test Hypotheses 3 and 4

As shown in Table 3, there is a statistically significant result dependent on gender for the intrinsic and subjective detox barrier or for the need to disconnect from digital devices and social media. To be specific, males show lower levels of disconnection from social media than females. Therefore, Hypothesis 4a is accepted. Nevertheless, there are no statistically significant results either for external and objective barriers to social media usage, or intrinsic and subjective or extrinsic and objective barriers to using devices. On this basis, Hypothesis 3a is rejected.

According to Table 4, age determines the difference due to external and objective barriers to using social media. Therefore, Hypothesis 3b, which proposes that the age of tourists explains external and objective barriers to using social media at the destination, is accepted. Besides this, the digital intrinsic and subjective detox barrier to using devices and the internet plays a role in behavior at the destination. On this basis, Hypothesis 4b, which states that the age of tourists explains intrinsic and subjective barriers is accepted.

Regarding the extrinsic and objective motives about infrastructural, social and time constraints to using social media at the destination, those who are younger suffer most from this inhibitor. On the other hand, older visitors are less likely to encounter this barrier. The need to disconnect from electronic devices and tourist social media networks is greater in people whose age is between 25 and 34 years, and also in tourists between 35 and 49, while the lowest intrinsic and subjective detox barrier appears in tourists between 18 and 24, as well as people over 65 years. In other words, middle-age is most closely associated with this negative detox motivation. Nevertheless, it is evident that age does not relate to either intrinsic and subjective apprehensive barriers or extrinsic and objective barriers for devices.

As shown in Table 5, nationality determines the differences in external barriers to social media usage and the intrinsic and subjective detox barrier, while not affecting differences in intrinsic and extrinsic barriers for devices. On that basis, Hypothesis 3c and 4c might be accepted. Therefore, the nationality of tourists explains differences in external and objective barriers to using social media at the destination. Similarly, the nationality of tourists explains intrinsic and subjective barriers, which might also be accepted.

Regarding the extrinsic and objective motives for using social media at the holiday destination, in relation to infrastructural, social and time constraints, Spanish and Dutch tourists reach the highest barrier levels, while German tourists report the lowest barrier levels. In terms of the digital detox holidays barrier, those who need to disconnect the most are tourists from Spain and “other countries”, while tourists from the Netherlands, the United Kingdom and Germany show the lowest levels of need to disconnect from devices and social media.

According to Table 6, education does not determine any statistically significant difference in external and objective barriers to social media usage, nor intrinsic and subjective detox barriers or intrinsic and extrinsic barriers to using devices. Hence, Hypotheses 3d and 4d, proposing that the education level of tourists explains external and objective barriers to using social media at the destination, but also subjective and intrinsic barriers are outright rejected.

As shown in Table 7, income determines the detox barrier to disconnecting from digital devices and social media but not the external and objective barriers to social media usage or the intrinsic and extrinsic barriers to using devices. The relationship between the digital intrinsic and subjective detox barrier and income is an inverse one, given that the higher the income, the less tourists feel the need to disconnect. For instance, tourists whose income is between >EUR 10,000 and <EUR 20,000 display a greater need to disconnect from digital devices and social media. On this basis, Hypothesis 4e, stating that the income of tourists explains intrinsic and subjective barriers to using social media at the destination, is accepted. Nevertheless, Hypothesis 3e is rejected, given that the income of tourists does not explain external or objective barriers.

## 5. Discussion and Conclusions

Although the concept of detox holidays seems to necessitate the avoidance of social media and digital devices [89], the destination experience is the sum of the visitors’ engagement with not only offline but also online dimensions [90]. For many authors, demystifying the essential authenticity of the non-digital reality is arguably as important a task as recognizing the potential and genuine influence of the online world [91]. Needless to say, both dimensions interplay and shape the visitor’s stay [92,93]. However, some tourists either do not bring their electronic devices or simply turn them off and refuse to connect to social media during their stay. For this reason, there is evidence that the existence of smart infrastructural destinations does not guarantee the development of the digital visitor experience. No doubt, technology is essential for enhancing the tourism experience [94,95,96]. Without e-commerce, GPS, augmented reality, and sharing on the spot, the destination experience would be diminished since these technologies are an essential part of the visit for some tourists [97]. Not only does technology shape the visiting experience [5,92], but it also implies a blended reality [91]. Therefore, it is vital that destination management highlights the barriers to using devices and connecting to social media. Similarly, describing the sociodemographic features—in the wider sense—of the impediments to going online at a destination seems to be a crucial factor. Why is it that some tourists are confronted with such a dilemma? That is, why do they sometimes refuse to participate in any digital activity on holiday? Not only is it essential for a destination’s success that this question is answered but it also represents a need for further research. What is more, as far as this dilemma is concerned, tourists question how healthy and beneficial digital systems are.

So far, the issue of technological access has represented the primary obstacle to enjoying a digital holiday and, hence, the predisposition toward using technology has been determined by the opportunities to use it. Nonetheless, several personal inhibitors determine the possibility of enjoying a wide-ranging holiday experience, besides those that account for the technologies available to visitors and their socioeconomic status. Thus, a digital experience at a destination entails acknowledging the tourist’s decision to have a digital holiday. What is more, the smarter the destination, the more the tourist’s decision emerges as the most significant hindrance to staying connected. On this basis, there is the question of why some tourists are reluctant to experience their holiday destination online. What inhibitors to going digital do they address? Is it due to perceived digital noise, the lack of a safe online environment, or health concerns?

The answer lies in recognizing the relationship between notions of work–life balance and a desire to disconnect [98], without eschewing the possibility of “e-mindfulness” by engaging in technology-assisted experiences [99,100,101]. No doubt, there is a digital ecosystem at a destination where the stakeholders come up with constant innovation in the form of e-commerce offers, co-creation activities, augmented reality, and so forth that are worth experiencing for the tourist [102]. Therefore, there must be a wiser approach than the approach known as “detox tourism”, such as, for example, “digital-free tourism”, moving to approaches where the strategy boils down to using technology in a more appropriate manner, reframing the ways of thinking and coping [103,104,105]. For example, it stands to reason that there is an increasing demand for public health initiatives to preserve tourists’ quality of life by resorting to inbound marketing policies.

This study pioneers the analysis of the detox barrier to staying connected on holiday and provides insight into how this intrinsic and subjective inhibitor interacts with other external hindrances. In other words, this research contribution essentially makes it clear that subjective barriers to using social media—that is, “digital-free travel’—are arguably as important as objective barriers—namely, the existence of “technology dead zones” [42]. For this reason, subjective and personal motivations appear alongside the quest for relationship improvements, besides the objective and external variables. In this vein, Hypotheses 1 and 2 were accepted. Going on the findings of Li et al. [106], there is a harmony between contextual, social, and personal factors, and Li et al. [107] indicate the paradoxical connection underlining the assumed disconnection. Likewise, although the sociodemographic profile that gives rise to the different digital behaviors has been quite extensively studied, the current paper sheds light on how gender, age, nationality, education, and income are associated with detox motives, which has so far been underexplored in the field of tourism. It is worth noting that not only are the subjective motives playing a vital role in the use of social media at the holiday destination but they are also enabling more sociodemographic diversity.

In particular, the current paper is an attempt to bring scholarly attention to the field of tourism, highlighting the fact that the differences in digital behaviors, at least in developed countries, offer a broader challenge to research than has been acknowledged so far. In addition, well-consolidated theoretical assumptions should be questioned. Firstly, one might claim that diminishing digital behavior at a holiday destination might be favorable for both tourists and resorts. In this respect, the obtained empirical evidence allows us to point out the existence of motivations that are chosen freely and driven by a desire to be disconnected—that is, digital-free tourism motives, such as escape, personal growth, health and well-being, and relationships [108]. For this reason, we might state that the digital behavior differences between home and holiday destination do not refer to inequality nor, necessarily, the digital divide [1]. In this way, the current paper upholds the possibility of exploring the different digital behavior between home and holiday destination from the perspective of health and wellness tourism, limiting the degree of intrusion, managing information and multiple tasks [62], and changing to new lifestyles [106] and ways of thinking [109]. There appear to be many theoretical approaches with public health implications that can be applied and extended in future research (see Figure 3).

To be specific, with the aim of measuring and segmenting all the existing degrees of digital difference, additional variables are relevant. The difference between having or not having devices, using or not using social media, and learning to use or ignoring literacy tools has given rise to a varied range of decision criteria on the part of tourists to remain intentionally disconnected. Clearly, different digital behaviors and their barriers more often than not relate to external and objective, infrastructural, social and time-based difficulties and constraints, although the obtained results also allow intrinsic and subjective, self-driven inhibitors to be highlighted. In this respect, it is worth mentioning some detox motives, the aims of which comprise alleviating stress and a desire for privacy and freedom, wellness and disconnection. In this sense, Hypothesis 2 was accepted. As far as new variables are found to be significantly related to the different digital behaviors, one might account for them by considering more psychological models of wellness [110], attitudes to “digital living” [111], and mental health [112]. Nevertheless, the intrinsic and subjective motives regarding concerns about losing or damaging one’s device during the holiday, as well as some apprehensions about being the victim of a robbery, burglary, or mugging, appear not to be associated with the different digital behaviors of tourists traveling between developed countries. So, although Hypothesis 2 was accepted, its acceptance is only partially demonstrated.

No doubt, insofar as new criteria for explaining the different digital behaviors in tourism have been revealed, practical implications might be suggested for the sake of optimal destination management. For example, further segmentation efforts should be made in order to provide for the emerging niche market of “detoxers” [113], whose treatment might be inspired by a health tourism management approach. For example, stress should be alleviated by offering outdoor activities without Wi-Fi intrusions [87,114,115]. Similarly, public policies should be implemented to research emerging health problems, prevent toxic online environments, and improve the quality of the user’s digital experiences.

Therefore, a new line of research is put forward in order to gain an understanding of a polymeric digital reality. Thus, to be comprehensive and exhaustive, while the health tourism doctrine should be included to analyze the different digital behaviors between home and the holiday destination for advanced countries, the criminal theories might be discarded as a cause of disconnection. Nevertheless, provided that a particular destination shows a lower level of security and socio-economic development, perhaps one might acknowledge the greater importance of these criminal theories.

In addition, it is important to point out that despite the more biological sociodemographic features, such as gender and age, being more strongly related to the different digital behaviors between home and holiday destination, the justifications tend to be sociological and psychological in nature. Thus, females and younger and older people are clearly impacted by the different digital behaviors between home and holiday destination, although this is due to different barriers and explanations. On the one hand, it stands to reason that women and the middle-aged show a greater propensity toward suffering from the causes that give rise to a need to detox, possibly because of their intense exposure to stressful digital experiences determined by different socialization patterns and roles, as well as “toxic” lifestyle circumstances [55]. Therefore, Hypothesis 4a was accepted. On the other hand, younger users face external and objective barriers related to infrastructural, social, and time-based constraints. Furthermore, although the older and younger subjects are the ones least inclined toward the detox motivation, it must be due to different reasons. While the former do not express this need, perhaps because of their lack of propensity for using digital devices, the latter are the most intensive users, and so must be more accustomed to new technologies. Thus, Hypotheses 3b and 4b were confirmed. In other words, there is a generational explanation that highlights the underlying effect of age on becoming digital “natives” or “immigrants” [64]. Nevertheless, the obtained evidence contradicts other studies [25,88,108] by pointing out that while younger people are more likely to be exposed to the downsides of technology and nomophobia, they are neither the most vulnerable, nor are they the most reluctant to go online on holiday.

Likewise, nationality seems to respond in accordance with the theory of digital “native” and “immigrant” countries, as well as “male” and “female” countries. To be specific, it is worth noting that “female” and “digital immigrant” countries, such as Spain, are more prone to suffering the causes that give rise to the need for digital detox than “male” and “digital native” countries, such as Germany and the UK. Nonetheless, the Netherlands, being a “female” culture, is free from detox motivations, possibly due to the fact that their tourists are the youngest. Consistently, this age variable accounts for the presence of an infrastructural, social, and time-based constraint explanation—despite the Netherlands being a digitally advanced country—since the younger tourists can rarely afford the best-quality services [1,15]. As was mentioned above, Hypotheses 3c and 4c were favorably contrasted.

Education does not show any relationship to the different digital behaviors between home and holiday destination, upholding the theory that knowledge and skills are not relevant, chiefly if tourists show similar levels of education when attempting to account for the lack of digital activity at the destination. Undoubtedly, Hypotheses 3d and 4d were rejected. Insofar as personal characteristics, for example, education, are not determining factors in explaining the adoption of innovations, this evidence is consistent with Roger’s diffusion theory of innovation since it gives credit to users’ requirements and personal choice [6]. In contrast, income shows an inverse relationship, although it must be determined by the kind of occupation in terms of the exposure to stress and the quality of life that the poorer subjects have, in comparison to the wealthier ones [86]. Thus, Hypothesis 4e was confirmed, even though Hypothesis 3e was rejected. In other words, this variable refers to inequality and even a digital divide [55] in which the less wealthy would be exposed to the worse consequences of “digital elasticity” [1,13].

On this basis, the current paper contributes to the literature by highlighting seemingly paradoxical evidence with a sociodemographic, theoretical consequence. It is worth noting that the biological variables of gender and age are more strongly related to the difference in digital behaviors than the sociotechnical variables stemming from education and skill. Needless to say, this evidence not only confirms the antecedent power of health tourism theories to account for the different digital behavior shown between home and holiday destination but also opens the door to describing how these biological features are shaped by culture in terms of roles and generational attributions, respectively.

Consistently, from the practitioner’s perspective, gender and age should be considered as the key criteria for the segmentation of the emerging market of digitally challenged tourists. It goes without saying that if female and middle-aged visitors are more likely to be affected by the causes that give rise to the need for a digital detox, due to their lifestyle circumstances, then they deserve appropriate treatment related to “slowdown” experiences and effective relaxation. Inbound marketing should be infused with this healthy policy and the public administration might take actions against any toxic digital influence, especially for women around the age of 40.

Finally, there emerge possible future lines of research that should be considered, in order to gain a much better understanding of digital-free tourism purposes. One of these refers to the evolution and potential changes of this free option during the stay. One might wonder whether the decision not to bring one’s device or to turn it off affects the entire holiday stay or, alternatively, if it evolves throughout the visit. If one’s mood causes a change in attitude toward using social media [116], the digital-free tourism purpose possibly changes too, does it not? Perhaps digital-free tourism might not be an absolute determination, but rather a passing holiday circumstance. Therefore, it would be worth researching the topic by distinguishing different phases in the travel and holiday periods. Secondly, as it seems that the type of tourism, for example, day trips, sightseeing tours, specialist tours, etc. might affect the use of social media, one might propose to look into digital detox behaviors in these contexts. Another future line of research might gear efforts toward measuring how the inherent mobility stress [117] interacts with digital detox, “unplugged” and digital-free tourism options. Finally, it seems advisable to draw on the existing theoretical framework that we have reviewed in this paper to develop and estimate empirical models, to explain tourist digital behaviors ranging from detox to e-mindfulness.

Last but not least, it is worth acknowledging that the present paper has several limitations. First, as these digital behaviors were measured using a questionnaire, there will be a gap between reported and actual conduct. Future research should use social media monitoring software and other web analytical tools to address this disparity. Second, it should be considered that the pre-stay behaviors were measured during the stay and this might bias the past through the lens of the present. To tackle this, prospective authors should survey longitudinally at both origin and destination. Third, other limitations stem from a non-probabilistic sampling procedure regarding destination representativeness.

## Figures and Tables

**Figure 1 ijerph-19-01548-f001:**
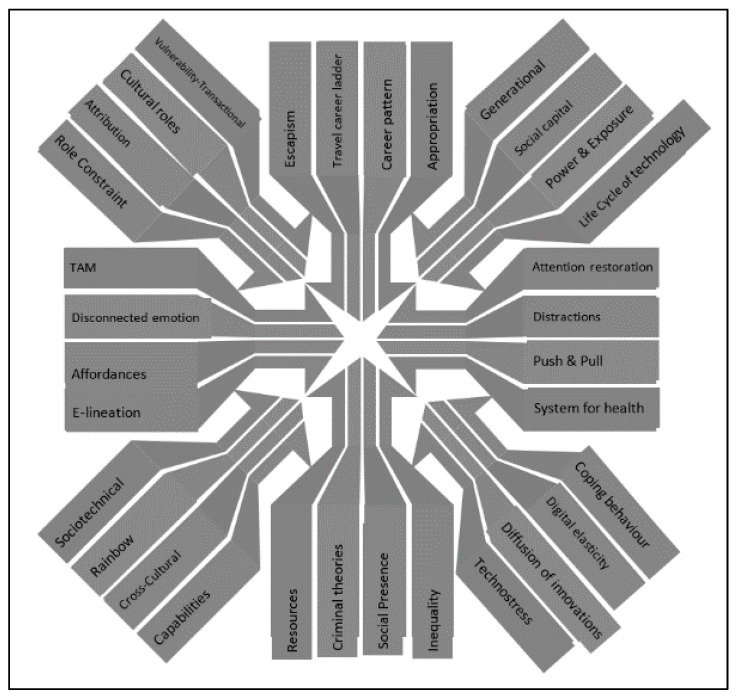
Theories and approaches to explaining tourists’ digital differences between destination and home.

**Figure 2 ijerph-19-01548-f002:**
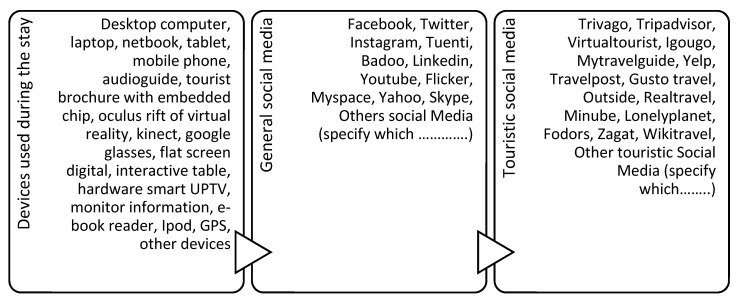
The questionnaire scales included devices, general social media, and touristic social media.

**Figure 3 ijerph-19-01548-f003:**
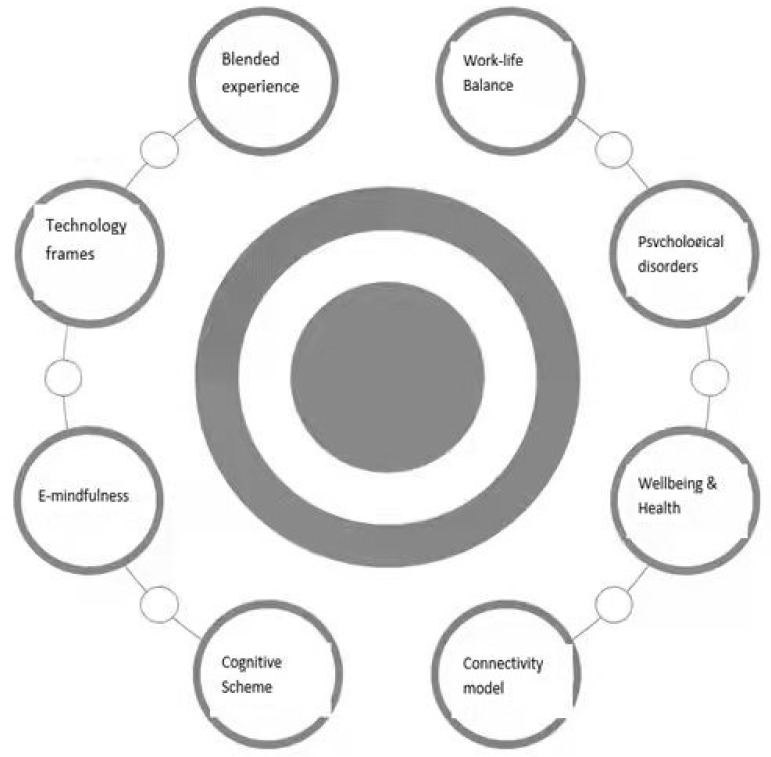
Theories and approaches to explain how to overcome visitors’ barriers to using social media at holiday destinations.

**Table 1 ijerph-19-01548-t001:** Exploratory factor analysis for the motivational barriers to connecting to social media and using devices at a holiday destination.

Comm.	Barrier Items	Rotated Matrix			
		F1	F2	F3	F4
0.710	Because I do not know where I can connect to social media	0.824	0.131	0.113	0.029
0.705	For technical failure or malfunction affecting social media	0.819	0.140	0.024	0.122
0.707	Because I have not had access to connecting devices and social media	0.801	0.096	0.051	0.231
0.616	Because I have not had access to the internet and social media	0.762	0.140	−0.076	0.100
0.678	Because my companions did not want me to connect to social media	0.650	0.074	0.499	0.023
0.450	Because I did not need social media	0.632	−0.148	0.146	0.082
0.554	Because I have not had time for social media	0.528	0.186	0.486	−0.065
0.804	Because my device could be stolen	0.120	0.872	0.113	0.132
0.810	Because I could lose my device	0.087	0.863	0.153	0.183
0.636	Because my device could break	0.111	0.717	0.289	0.162
0.600	Because I needed to unwind and relax away from my device	−0.042	0.288	0.699	0.162
0.676	Because my companions did not want me to bring my device	0.107	0.115	0.635	0.498
0.670	Because I wanted a break from social media	0.491	0.278	0.586	−0.090
0.594	For the technological incompatibility of my device with the destination	0.038	0.068	0.154	0.751
0.538	Because the device was broken	0.124	0.145	0.214	0.675
0.359	Because I wanted to avoid excess baggage due to my device	0.136	0.179	−0.172	0.528
KMO: 0.840; Bartlett: 2358.136; degree of freedom: 120; sig. 0.000, explained variance: 63.158%					

**Table 2 ijerph-19-01548-t002:** Correlation analysis between devices, general social media and tourist social media differences, and the motivational barriers to connecting at the holiday destination.

		F1	F2	F3	F4
Device difference	C. Pearson	0.122 *	0.092	0.068	0.164 **
	Sig. (bilateral)	0.024	0.089	0.204	0.002
	N	346	346	346	346
General social media difference	C. Pearson	0.436 **	0.096	0.079	0.057
	Sig. (bilateral)	0.000	0.076	0.145	0.287
	N	346	346	346	346
Tourist social media difference	C. Pearson	0.271 **	0.085	0.228 **	0.068
	Sig. (bilateral)	0.000	0.116	0.000	0.209
	N	346	346	346	346

* Significance 0.05, ** Significance 0.01.

**Table 3 ijerph-19-01548-t003:** Student’s *t*-test difference of means: home versus destination levels of barriers, depending on gender.

Levene’s Test for Equality of Variances				*t*-Test for Equality of Means				
		F	Sig.	*t*	Df	Sig. (2-Tailed)	Mean Difference	Std. Error Difference
F1	Equal variances	0.824	0.365	−1.125	344	0.261	−0.12151727	0.10800196
	Not Equal variances			−1.126	332.372	0.261	−0.12151727	0.10788418
F2	Equal variances	2.203	0.139	0.775	344	0.439	0.08381891	0.10810608
	Not Equal variances			0.781	338.488	0.436	0.08381891	0.10735953
F3	Equal variances	5.419	0.020	−0.705	344	0.481	−0.07621010	0.10812245
	Not Equal variances			−0.689	292.501	0.491	−0.07621010	0.11057535
F4	Equal variances	0.002	0.963	−0.358	344	0.721	−0.03871799	0.10818036
	Not Equal variances			−0.357	326.504	0.722	−0.03871799	0.10854010
Group Statistics								
Gender		N	Mean		Stand. Deviat.		Stand. Deviat. mean	
F1	Male	190	−0.0547881		1.00458380		0.07288017	
	Female	156	0.0667291		0.99352220		0.07954544	
F2	Male	190	0.0377912		1.03158083		0.07483874	
	Female	156	−0.0460277		0.96142402		0.07697553	
F3	Male	190	−0.0343606		0.88958214		0.06453707	
	Female	156	0.0418495		1.12145097		0.08978794	
F4	Male	190	−0.0174567		0.98589981		0.07152469	
	Female	156	0.0212613		1.01968963		0.08164051	

**Table 4 ijerph-19-01548-t004:** ANOVA analysis to test the difference of given barriers, depending on age.

ANOVA						
		Sum of Squares	Df	Mean Square	F	Sig.
F1	Between Groups	28.082	4	7.021	7.562	0.000
	Within Groups	314.706	339	0.928		
	Total	342.788	343			
F2	Between Groups	1.565	4	0.391	0.387	0.818
	Within Groups	342.564	339	1.011		
	Total	344.130	343			
F3	Between Groups	14.071	4	3.518	3.618	0.007
	Within Groups	329.590	339	0.972		
	Total	343.661	343			
F4	Between Groups	0.460	4	0.115	0.114	0.977
	Within Groups	340.927	339	1.006		
	Total	341.387	343			
Descriptive						
		N	Mean	Std. Deviation	Std. Error	
F1	18–24 years	14	0.5959056	1.38137611	0.36918830	
	25–34 years	50	0.3225656	0.57796993	0.08173729	
	35–49 years	68	0.1910295	0.77934229	0.09450913	
	50–64 years	128	−0.0031781	1.02630345	0.09071327	
	65–100 years	84	−0.4285727	1.09251355	0.11920300	
	Total	344	0.0030641	0.99969055	0.05389970	
F2	18–24 years	14	0.0216820	1.17705319	0.31458070	
	25–34 years	50	0.1419495	1.17386785	0.16600998	
	35–49 years	68	0.0451392	1.06022972	0.12857174	
	50–64 years	128	−0.0357547	0.87508384	0.07734721	
	65–100 years	84	−0.0549817	1.00879509	0.11006857	
	Total	344	0.0037076	1.00164577	0.05400512	
F3	18–24 years	14	−0.2184192	1.04904517	0.28036911	
	25–34 years	50	0.4306319	1.20409834	0.17028522	
	35–49 years	68	0.0170299	0.92843601	0.11258940	
	50–64 years	128	0.0045554	1.01186578	0.08943714	
	65–100 years	84	−0.2213592	0.82473688	0.08998617	
	Total	344	0.0047113	1.00096346	0.05396833	
F4	18–24 years	14	−0.0586405	0.87934746	0.23501549	
	25–34 years	50	−0.0449929	1.04886655	0.14833213	
	35–49 years	68	−0.0132354	0.83771970	0.10158844	
	50–64 years	128	0.0014051	0.88688852	0.07839061	
	65–100 years	84	0.0600238	1.25177260	0.13657959	
	Total	344	0.0036373	0.99764566	0.05378945	

**Table 5 ijerph-19-01548-t005:** ANOVA analysis to test the differences in the given barriers, depending on nationality.

ANOVA							
		Sum of Squares		Df	Mean Square	F	Sig.
F1	Between Groups	19.292		5	3.858	4.028	0.001
	Within Groups	325.708		340	0.958		
	Total	345.000		345			
F2	Between Groups	9.152		5	1.830	1.853	0.102
	Within Groups	335.848		340	0.988		
	Total	345.000		345			
F3	Between Groups	19.190		5	3.838	4.005	0.002
	Within Groups	325.810		340	0.958		
	Total	345.000		345			
F4	Between Groups	6.784		5	1.357	1.364	0.237
	Within Groups	338.216		340	0.995		
	Total	345.000		345			
Descriptive							
		N	Mean	Std. Deviation		Std. Error	
F1	Germany	68	−0.4125464	1.30696894		0.15849326	
	United Kingdom	50	0.0077219	0.79948692		0.11306452	
	Netherlands	24	0.0536534	1.15892007		0.23656357	
	Nordic countries	88	0.0074064	0.99797213		0.10638419	
	Spain	85	0.2997930	0.69602323		0.07549432	
	Others	31	0.0079099	0.85066815		0.15278451	
	Total	346	0.0000000	1.00000000		0.05376033	
F2	Germany	68	−0.1250788	0.77253186		0.09368325	
	United Kingdom	50	−0.1409838	0.80366791		0.11365581	
	Netherlands	24	0.1879608	0.98129651		0.20030631	
	Nordic countries	88	0.0250333	1.00637864		0.10728032	
	Spain	85	0.2079239	1.23017899		0.13343165	
	Others	31	−0.2849348	0.94572421		0.16985708	
	Total	346	0.0000000	1.00000000		0.05376033	
F3	Germany	68	−0.0804274	0.78627951		0.09535040	
	United Kingdom	50	−0.1648697	0.74939815		0.10598090	
	Netherlands	24	−0.3965603	0.61978531		0.12651315	
	Nordic countries	88	−0.1166319	0.93530671		0.09970403	
	Spain	85	0.3816987	1.33563789		0.14487027	
	Others	31	0.0338457	0.85785825		0.15407589	
	Total	346	0.0000000	1.00000000		0.05376033	
F4	Germany	68	0.0387311	1.12153307		0.13600586	
	United Kingdom	50	−0.1391786	0.77196253		0.10917199	
	Netherlands	24	0.2563112	1.04100860		0.21249499	
	Nordic countries	88	0.0103806	1.00367501		0.10699212	
	Spain	85	−0.1366409	0.73743204		0.07998573	
	Others	31	0.2862819	1.48878888		0.26739438	
	Total	346	0.0000000	1.00000000		0.05376033	

**Table 6 ijerph-19-01548-t006:** ANOVA analysis to test the difference of given barriers depending on education.

ANOVA						
		Sum of Squares	Df	Mean Square	F	Sig.
F1	Between Groups	0.573	4	0.143	0.143	0.966
	Within Groups	335.442	335	1.001		
	Total	336.014	339			
F2	Between Groups	1.734	4	0.433	0.427	0.789
	Within Groups	340.055	335	1.015		
	Total	341.788	339			
F3	Between Groups	7.169	4	1.792	1.801	0.128
	Within Groups	333.278	335	0.995		
	Total	340.447	339			
F4	Between Groups	3.978	4	0.995	1.001	0.407
	Within Groups	332.823	335	0.994		
	Total	336.801	339			
Descriptive						
		N	Mean	Std. Deviation	Std. Error	
F1	Without studies	1	−0.0581295			
	Primaries	32	−0.0519640	1.15924865	0.20492815	
	High School/Professional	56	−0.0396226	0.96424751	0.12885299	
	Bachelor—University	123	0.0004032	1.03627408	0.09343769	
	Master’s/Doctorate	128	0.0580852	0.93755062	0.08286855	
	Total	340	0.0104254	0.99558656	0.05399326	
F2	Without studies	1	−0.2918616			
	Primaries	32	0.0610274	1.05109794	0.18580962	
	High School/Professional	56	0.1563399	1.17343040	0.15680623	
	Bachelor—University	123	−0.0129281	1.01470853	0.09149319	
	Master/Doctorate	128	−0.0381171	0.90692262	0.08016139	
	Total	340	0.0116085	1.00410420	0.05445520	
F3	Without studies	1	−0.3976406			
	Primaries	32	0.4194802	1.32450569	0.23414174	
	High School/Professional	56	0.0683781	1.01401864	0.13550394	
	Bachelor—University	123	−0.0958073	0.93369582	0.08418852	
	Master/Doctorate	128	−0.0132330	0.95564496	0.08446788	
	Total	340	0.0099317	1.00213212	0.05434824	
F4	Without studies	1	−0.3344006			
	Primaries	32	−0.0091609	0.84987967	0.15023892	
	High School/Professional	56	0.0608089	0.93887209	0.12546206	
	Bachelor—University	123	−0.1174149	0.77635386	0.07000147	
	Master’s/Doctorate	128	0.1255723	1.21803274	0.10765990	
	Total	340	0.0129676	0.99675172	0.05405645	

**Table 7 ijerph-19-01548-t007:** ANOVA analysis to test the differences of the given barriers, depending on income.

ANOVA							
		Sum of Squares		Df	Mean Square	F	Sig.
F1	Between Groups	6.302		3	2.101	2.039	0.109
	Within Groups	303.965		295	1.030		
	Total	310.267		298			
F2	Between Groups	2.693		3	0.898	0.847	0.469
	Within Groups	312.783		295	1.060		
	Total	315.477		298			
F3	Between Groups	10.419		3	3.473	3.440	0.017
	Within Groups	297.816		295	1.010		
	Total	308.235		298			
F4	Between Groups	2.934		3	0.978	1.056	0.368
	Within Groups	273.076		295	0.926		
	Total	276.010		298			
Descriptive							
Income in EUR		N	Mean	Std. Deviation		Std. Error	
F1	>10,000 <20,000	92	0.1731977	0.93351356		0.09732552	
	>20,000 <40,000	82	−0.0100999	1.13362825		0.12518830	
	>40,000 <100,000	99	−0.1609455	0.96010358		0.09649404	
	>100,000 <400,000	26	0.1992638	1.09969586		0.21566810	
	Total	299	0.0145594	1.02037465		0.05900979	
F2	>10,000 <20,000	92	0.1653728	1.18506148		0.12355120	
	>20,000 <40,000	82	−0.0541992	0.86833506		0.09589157	
	>40,000 <100,000	99	0.0199483	1.04789414		0.10531732	
	>100,000 <400,000	26	−0.0972929	0.80743719		0.15835146	
	Total	299	0.0341646	1.02890530		0.05950313	
F3	>10,000 <20,000	92	0.2908242	1.25511086		0.13085435	
	>20,000 <40,000	82	−0.1221184	0.86450298		0.09546838	
	>40,000 <100,000	99	−0.0311943	0.86557371		0.08699343	
	>100,000 <400,000	26	−0.2403379	0.90561751		0.17760621	
	Total	299	0.0247662	1.01702781		0.05881624	
F4	>10,000 <20,000	92	−0.0820250	0.75946799		0.07918001	
	>20,000 <40,000	82	−0.0310082	0.79015803		0.08725836	
	>40,000 <100,000	99	0.1378265	1.24529572		0.12515693	
	>100,000 <400,000	26	−0.1170060	0.84950284		0.16660121	
	Total	299	0.0017181	0.96239682		0.05565685	

## Data Availability

Database is available if required.

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
