# Peer review of "The Outbreak of Digital Detox Motives and Their Public Health Implications for Holiday Destinations"

_ijerph, 2022, doi:10.3390/ijerph19031548_

Round 1

Reviewer 1 Report

Strengths

  1. The theme is relevant and has some degree of novelty.
  2. The manuscript has a proper structure and is well written.
  3. The hypotheses are carefully supported in the literature.
  4. The methodology and statistical treatment of data are adequate

Suggestions for improvement

  1. Questions referring to measure of devices, measure of generic social media, and measure of touristic social media should be presented in the manuscript.
  2. The source of the scale measuring the motivational barriers to connecting to social media and using devices at destination should be indicated.
  3. On line 442, where is “H2d” it should be “H2e”.
  4. In the discussing of results, I suggest explicitly indicating the hypothesis that is being discussed in each part of this chapter.
  5. Study limitations are missing.

Author Response

Response to reviewer 1

We feel grateful for your review and comments and for highlighting our paper’s strengths regarding the theme, structure, hypotheses support and methodology. In addition, we thank your suggestions to improve and provide you with detailed answers as follows:

REVIEWER 2 SUGGESTION

AUTHORS’ RESPONSES

1

Questions referring to measure of devices, measure of generic social media, and measure of touristic social media should be presented in the manuscript.

Thank you for this suggestion. We have inserted a new figure 2 in which we display the questionnaire’s items regarding the devices, general social media and touristic social media.

2

The source of the scale measuring the motivational barriers to connecting to social media and using devices at the destination should be indicated.

The sources of this scale are now disclosed

3

On line 442, where is “H2d” it should be “H2e”.

Sorry. The mistake has been amended

4

In the discussing of results, I suggest explicitly indicating the hypothesis that is being discussed in each part of this chapter.

All the hypotheses are now mentioned in the discussion section

5

Study limitations are missing.

We have appended some limitations to the paper as follows:

Reviewer 2 Report

This research study is interesting for me and deserves to be published. The study contributes to the limited studies of the intrinsic and subjective inhibitors to remaining online at a touristic destination.

However, there are several points of concern.

  1.  

My main objection concerns the correctness of formulating the hypotheses - their complexity is a mistake. I propose to divide H1 (line 150-151) into two hypotheses:

H1: The different digital responses of tourists between origin and destination is associated with external and objective barriers.

H2: The different digital responses of tourists between origin and destination is associated with intrinsic and subjective barriers.

Similarly, I propose to divide H2 (line 167-168) into hypotheses:

H3 The sociodemographic profile of tourists explains external and objective barriers to going online at destination.

H4 The sociodemographic profile of tourists explains intrinsic and subjective barriers to going online at destination.

A similar procedure would be necessary for the current H2a-e hypotheses.

2.

Furthermore, I propose to consider whether in the hypotheses and throughout the article it would not be more appropriate to use the term "digital behavior" instead of "digital responses".

3.

As the authors focus their research on the use of social media and not the Internet in general, the term "using social media" instead of "going online" should be used in the hypotheses.

The Internet can be used for various purposes, also unrelated to social media (e.g. information, entertainment). Tourists can disconnect themselves from social media and still use google maps or Trip Advisor sites, for example, without seeing them as an obstacle to "escape, relaxation and well-being".

4.

490-492: ”In other words, this research contribution essentially makes it clear that subjective barriers to going online - that is, ‘digital free travel’ - are more significant than objective barriers - namely, the existence of ‘technology dead zones’.”

and

500-501: “It is worth noting that not only are the subjective motives more relevant, but they also carry more sociodemographic diversity”. 

- This statement is inconsistent with the results of the study – line 378-379

5.

547 and 552

Instead of "criminal theory" it should be "criminal theories".

6.

505-521

This is a generally hard-to-understand passage:

“Firstly, one might question whether the existence of a different digital response between home and destination is always negative (Incomprehensible, why such a question arises - maybe it is worth indicating in what context or from whose point of view? Tourist? Destination places?). In this respect, the obtained empirical evidence allows us to point out the existence of motivations that are chosen freely and driven by a desire to be disconnected - that is, digital free tourism motives such as escape, personal growth, health and well-being, and relationships [106]. Eclectically (?), the obtained result is also consistent with [1], who indicated that digital connection might sometimes be negative in nature. Nevertheless, we must assert that whilst the difference is always positive (in what sense? Why?), absolute disconnection is negative insofar as tourists might miss out all the potential experiences the destination has to offer. On this basis, although any difference in digital responses at destination seems to refer to a categorical and total inequality (?), one might propose a more linear and relative difference (?) whose complex reality might be accounted for by new polymeric (?) theoretical approaches beyond the traditional doctrines stemming from sociotechnical frameworks (?). In this way, the current paper upholds the possibility of looking into the different digital response between home and destination from the perspective of health and wellness tourism, limiting the degree of intrusion, managing information and multiple tasks [62] and changing into new lifestyles [104] and ways of thinking [107]. In this vein, there appear to be many theoretical approaches that can be applied and extended in future research (see figure 2).”

7.

609 et seq

It seems that the type of tourism (e.g. stay trips, sightseeing tours, specialist tours) will affect the use of the Internet, so it may also be a further direction of research.

8.

Where there are direct references to other authors in the article, their names and years should also appear, and not only a numerical reference to the sources (e.g. line 511, 544-555) because it makes reading very difficult.

9.

I would propose the abandonment of colors in Figs. 1 and 2, because they suggest the existence of some key for assigning them to particular theories (and I don't think it is there).

Author Response

Response to reviewer 2

We would like to thank you for your review and comments and for considering that the study is interesting and deserving publication. Besides, we feel grateful for remarking that there are a limited number of studies on our topic. Please, see below our specific answers to each of your comments.

REVIEWER 2 SUGGESTION

AUTHORS’ RESPONSES

1

My main objection concerns the correctness of formulating the hypotheses - their complexity is a mistake. I propose to divide H1 (line 150-151) into two hypotheses:

H1: The different digital responses of tourists between origin and destination is associated with external and objective barriers.

H2: The different digital responses of tourists between origin and destination is associated with intrinsic and subjective barriers.

Similarly, I propose to divide H2 (line 167-168) into hypotheses:

H3 The sociodemographic profile of tourists explains external and objective barriers to going online at destination.

H4 The sociodemographic profile of tourists explains intrinsic and subjective barriers to going online at destination.

A similar procedure would be necessary for the current H2a-e hypotheses.

Thank you for this suggestion. We have revised the hypotheses as you did and followed your indications for former H2a-e. Consequently, the final version of the hypotheses can be found in the review of the literature and the analysis of the results sections.

2

Furthermore, I propose to consider whether in the hypotheses and throughout the article it would not be more appropriate to use the term "digital behavior" instead of "digital responses".

Following your suggestion, we have replaced “digital responses” with “digital behaviours”

3

As the authors focus their research on the use of social media and not the Internet in general, the term "using social media" instead of "going online" should be used in the hypotheses.

The Internet can be used for various purposes, also unrelated to social media (e.g. information, entertainment). Tourists can disconnect themselves from social media and still use google maps or Trip Advisor sites, for example, without seeing them as an obstacle to "escape, relaxation and well-being".

Following your indication, we have substituted the expression “going online” for “using social media.”

4

490-492: ”In other words, this research contribution essentially makes it clear that subjective barriers to going online - that is, ‘digital free travel’ - are more significant than objective barriers - namely, the existence of ‘technology dead zones’.”

and

500-501: “It is worth noting that not only are the subjective motives more relevant, but they also carry more sociodemographic diversity”. 

- This statement is inconsistent with the results of the study – line 378-379

Thank you. We have gone back over the inconsistent paragraph to work it out with coherency.

Round 2

Reviewer 2 Report

Thank you very much to the Authors for taking into account my suggestions. I have no further comments on the article.